# SCOPE enables type III CRISPR-Cas diagnostics using flexible targeting and stringent CARF ribonuclease activation

Jurre A. Steens[1,2,13], Yifan Zhu[1,13 ✉], David W. Taylor [3,4,5,6], Jack P. K. Bravo[3], Stijn H. P. Prinsen [2], Cor D. Schoen[7], Bart J. F. Keijser[8], Michel Ossendrijver[8], L. Marije Hofstra[9], Stan J. J. Brouns [10], Akeo Shinkai [11,12], John van der Oost [1] & Raymond H. J. Staals [1 ✉]

Characteristic properties of type III CRISPR-Cas systems include recognition of target RNA and the subsequent induction of a multifaceted immune response. This involves sequence-specific cleavage of the target RNA and production of cyclic oligoadenylate (cOA) molecules. Here we report that an exposed seed region at the 3′ end of the crRNA is essential for target RNA binding and cleavage, whereas cOA production requires base pairing at the 5′ end of the crRNA. Moreover, we uncover that the variation in the size and composition of type III complexes within a single host results in variable seed regions. This may prevent escape by invading genetic elements, while controlling cOA production tightly to prevent unnecessary damage to the host. Lastly, we use these findings to develop a new diagnostic tool, SCOPE, for the specific detection of SARS-CoV-2 from human nasal swab samples, revealing sensitivities in the atto-molar range.

[1] Laboratory of Microbiology, Wageningen University and Research, Wageningen, The Netherlands. [2] Scope Biosciences, Wageningen, The Netherlands. [3] Department of Molecular Biosciences, University of Texas at Austin, Austin, TX, USA. [4] Institute for Cellular and Molecular Biology, University of Texas at Austin, Austin, TX, USA. [5] Center for Systems and Synthetic Biology, University of Texas at Austin, Austin, TX, USA. [6] LIVESTRONG Cancer Institutes, Dell Medical School, Austin, TX, USA. [7] BioInteractions and Plant Health, Wageningen Plant Research, Wageningen, The Netherlands. [8] TNO, Zeist, The Netherlands. [9] Virology, Department of Medical Microbiology, University Medical Center Utrecht, Utrecht, The Netherlands. [10] Department of Bionanoscience, Delft University of Technology, Delft, The Netherlands. [11] RIKEN SPring-8 Center, Sayo, Hyogo, Japan. [12] Present address: RIKEN Cluster for Pioneering Research, Wako, Saitama, Japan. [13] These authors contributed equally: Jurre A. Steens, Yifan Zhu. ✉email: zhuyi_fan@aliyun.com; Raymond.Staals@wur.nl

As a widespread prokaryotic adaptive immune system, CRISPR-Cas (Clustered Regularly Interspaced Short Palindromic Repeats/CRISPR-associated) systems target and cleave genetic material of viruses and other mobile genetic elements (MGEs)[1–4]. A CRISPR array is composed of alternating repeat and spacer sequences, typically with the repeats consisting of identical sequences and the spacers consisting of variable sequence fragments acquired from invading MGEs[5–7]. The CRISPR array is generally located adjacent to a set of CRISPR-associated (cas) genes encoding the Cas proteins.

In type III CRISPR-Cas systems, the CRISPR array is expressed and processed into mature CRISPR RNA (crRNA) by the Cas6 ribonuclease[8]. A second maturation event occurs where the crRNA is trimmed at the 3′ end[9,10]. The mature crRNAs form a ribonucleoprotein complex together with a set of Cas proteins: the type III effector complex (Fig. 1a). In the interference stage,

type III CRISPR-Cas systems are unique in that they attack nucleic acids in three distinct ways.

Unlike other CRISPR-Cas systems that exclusively target either DNA (type I, II and V) or RNA (type VI), many type III systems have the capacity to target both RNA and DNA[11–17]. Structural and biochemical analyses of the multi-subunit type III complexes have revealed the subunits responsible for these activities and showed that they are induced in a step-wise manner, starting with the binding of a complementary target RNA[15,18–23]. After binding, the target RNA is cleaved by the Cas7 ribonuclease, which is present in multiple copies and constitutes the backbone of type III complexes. As such, type III interference complexes have multiple (2–4) active sites, cleaving the target RNA at 6 nt intervals (Fig. 1a)[12,15,24–26]. Simultaneously, target RNA binding activates two distinct catalytic domains of the Cas10 protein, the large subunit of type III complexes. Activation of the HD domain of

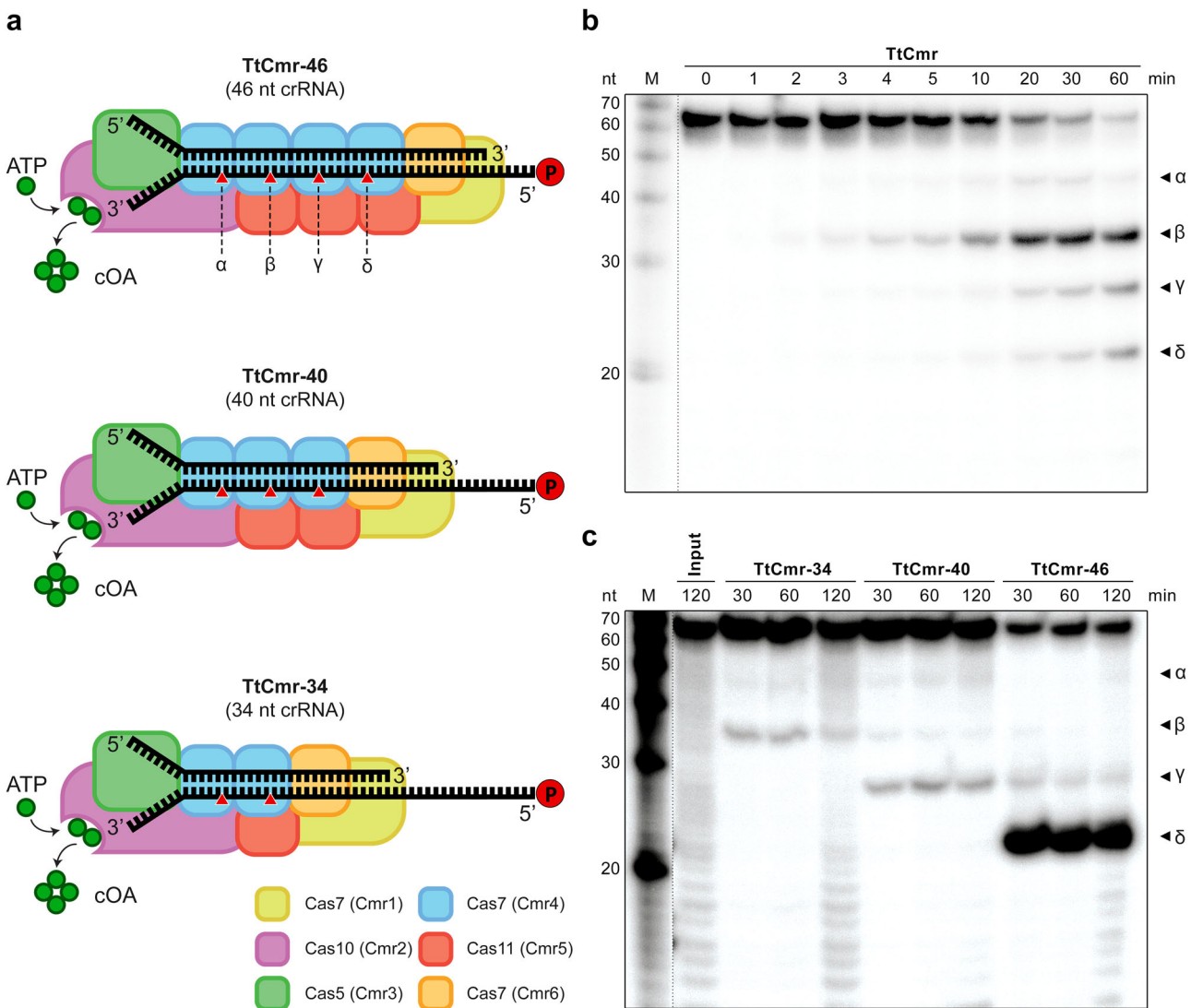

**Fig. 1 In vitro RNase activity assays with the endogenous and reconstituted TtCmr complexes. a** Schematic illustration of the different reconstituted TtCmr complexes used in the activity assays shown in panel **c**, pre-loaded with either the 46 (TtCmr-46), 40 (TtCmr-40) or 34 nt (TtCmr-34) crRNA (top strand). Red triangles indicate the anticipated cleavage sites (α, β, γ and δ) in the 4.5 target RNA (bottom strand, Supplementary Table S1) by the endoribonuclease activity of the Cas7 subunits. The target RNA was radiolabeled at the 5′ end with $^{32}$P γ-ATP ("P" in the red circle). **b** Denaturing PAGE analysis of the activity assay using a 5′ labeled target RNA complementary to the crRNA incubated with the endogenous TtCmr complex. A single stranded RNA marker ("M") was used as size standards as indicated on the left. **c** Activity assays similar to panel **b** but using the reconstituted complexes. Discontinuous gel lanes are indicated by a dashed line. The results of the cleavage assays are representative results of three (**b**) or two (**c**) replicates (Supplementary Fig. 6). Source data are provided as a Source data file.

Cas10 confers sequence-nonspecific DNase activity[16,26,27], while activation of its Palm domain triggers oligoadenylate cyclase activity, producing cyclic oligoadenylate (cOA) second messenger molecules that allosterically activate CARF (CRISPR-associated Rossmann fold) proteins[27,28]. Most of the CARF proteins that have been characterized so far appear to be promiscuous RNases (fusion of CARF and HEPN domains), cleaving both viral and host RNAs, thereby potentially inducing cell dormancy or cell death[29–31].

Previously, we characterized the structural and enzymatic features of the endogenous type III-B Cmr complex from *T. thermophilus* HB8 (TtCmr)[23,32]. We showed that TtCmr adopts a structure similar to type I (Cascade) complexes: a backbone consisting of several Cas7 (Cmr4) subunits, associated with multiple copies of the small subunit Cas11 (Cmr5). The complex is capped at one end by a heterodimer of the large subunit Cas10 (Cmr2) and Cas5 (Cmr3), and at the other end by a heterodimer of Cas7-like subunits (Cmr1 and Cmr6)[23]. It is important to note that the Cas10 subunit of the TtCmr complex lacks the HD domain, and hence does not have DNase activity[18]. The mature crRNA runs along the Cas7 backbone of the complex, with its 5′ repeat-derived end (the 5′ handle) anchored by Cas10/Cas5, and its 3′ end located at the Cmr1/Cmr6 end[20,23].

Interestingly, the 3′ end of the mature crRNA is variable in type III systems, due to an uncharacterized 3′ processing event following the endonucleolytic cleavages of Cas6[33–35]. Although the details of this 3′ processing event are not known, it is hypothesized that the heterogenous nature of the TtCmr complex might be responsible for this. Indeed, analysis of the crRNA-content of the endogenous TtCmr complex showed that it indeed co-purifies with mature crRNAs of different sizes (with variable 3′ ends), with a distinct 6-nt pattern: 34, 40 & 46 nt[18]. In addition, our previously obtained cryo-EM structures revealed that the native population of TtCmr complexes consisted of larger (with a stoichiometry of $Cmr1_21_31_44_53_61$) and smaller complexes (i.e., $Cmr1_21_31_43_52_61$ and $Cmr1_21_31_42_51_61$), with the smaller complexes lacking one or two Cas7–Cas11 (Cmr4-Cmr5) backbone segment(s) (Fig. 1a). Taken together, these data indicate that the 3′ end of the mature crRNA is determined by the stoichiometry of the TtCmr complex. In this scenario, it is likely that the 3′ end is generated by a (non-Cas) host ribonuclease, that shortens the unprotected, protruding 3′ end of the bound crRNA[35].

In this work, we set out to understand the biological significance of these differently-sized Cmr complexes. We reveal yet another unique feature of type III: a flexible seed region at the 3′ end of the crRNA guides, that appears to be important for these systems to prevent phage escapees. Additionally, we identify another key feature at the 5′ end of the type III crRNA that, upon binding a perfectly match target sequence, triggers the catalytic activities of Cas10, thereby ensuring tight control over CARF protein activation. These characteristics form the basis for the development of a highly sensitive type III diagnostics platform called SCOPE (Screening using CRISPR Oligoadenylate-Perceptive Effectors).

## Results

### Size variation of TtCmr complexes

We previously demonstrated that endogenous type III-B Cmr complexes purified from *T. thermophilus* HB8 (TtCmr) are loaded with mature crRNA guides of different lengths (34-40-46 nt), and that the crRNA-4.5 (CRISPR array 4, spacer 5) is most abundant (Fig. 1a)[18]. This endogenous Cmr complex specifically cleaves complementary target RNAs (4.5 target RNA) at 6 nt intervals, corresponding to the Cas7 subunits in the backbone of the complex. Consequently, this results in 5′ labeled degradation products of 39 (α), 33 (β),

27 (γ) and 21 (δ) nucleotides (Fig. 1b). However, the heterogeneous nature of the crRNA-content of the endogenous Cmr complexes[18,23], complicates the interpretation of these results. Therefore, to further reveal the mechanism of target RNA cleavage, we used *E. coli*-produced subunits to reconstitute three different Cmr complexes bound to a single crRNA (crRNA-4.5) of a defined length. Based on their abundance in their native host[18], we chose to include crRNA lengths of either 34 (TtCmr-34), 40 (TtCmr-40) or 46 nt (TtCmr-46). Opposed to all four (α-δ) 5′-labeled degradation products observed with the endogenous complex, each of the reconstituted complexes produced defined degradation products decreasing in size with a longer crRNA (Fig. 1c). This is consistent with the idea that the composition of the complex corresponds to the length of the crRNA, with larger complexes (e.g., TtCmr-46) harboring more cleavage sites, hence cleaving more closely to the labeled 5′ end of the target RNA. Smaller complexes, such as TtCmr-40 and TtCmr-34, lack one or two Cas7–Cas11 (Cmr4-Cmr5) backbone segment(s), respectively, and therefore cleave the target RNA at less and more distal locations (further away from the 5′ label), resulting in larger degradation products. These results show that the population of endogenous TtCmr complexes is a heterogenous mixture of bigger and smaller complexes, cleaving their cognate target RNAs at different positions.

### Flexible 3′ seed region

To investigate the significance of these type III complexes with different stoichiometries, we performed activity assays to probe for differences in seed requirements for RNA targeting as well as for the production of cyclic oligoadenylate (cOA) second messengers. In the structurally-related type I effector complexes (i.e., the Cascade complex), DNA targeting is governed by two factors: the PAM (protospacer adjacent motif) and the seed[36–38]. In the RNA-targeting type III systems, however, self/non-self discrimination is conferred by an rPAM (RNA protospacer-adjacent motif)[14]. This motif checks for complementarity between the 5′ handle of the crRNA (8 nucleotides, referred to as nucleotides −8 to −1) and the corresponding 3′ region flanking the protospacer[15,16,39,40] (Fig. 2a). Since TtCmr is devoid of DNase activity[18,23] we tested whether RNase activity and production of cOA are affected by target RNAs with complementarity to the 5′ handle of the crRNA (Fig. 2a). The cleavage activity assays with the endogenous Cmr complex showed that these 'self-like' substrates had no substantial effect on RNA cleavage activity (Fig. 2b). To probe for their impact on cOA production, we developed a pyrophosphatase-based assay that directly reflects the oligoadenylate oligomerization, as a measure for cOA production. A by-product of cOA production is the formation of pyrophosphate (PPi), which can be converted to free phosphate (Pi) by a thermostable pyrophosphatase enzyme[27] and be subsequently visualized by the Malachite Green colorimetric technique. Using this assay, we show that target RNAs with complementarity to the 5′ handle reduced the production of cOA to background levels, comparable to using a non-target (NT) target RNA (Fig. 2c). Similar results were obtained for the reconstituted TtCmr-46 and TtCmr-40 complexes (Supplementary Fig. S1). We conclude that cOA production, but not target RNA cleavage, is affected by complementarity between target RNA and the 5′ handle of the crRNA, thereby mitigating detrimental consequences that cOA production might cause (e.g., cell death/dormancy) when binding antisense transcripts from the CRISPR array[12,13].

Next, we tested whether TtCmr utilizes a seed similar as described for the Cascade complexes of type I systems. Activity assays were performed by incubating the TtCmr complex with target RNAs containing single mismatch mutations in the first 7

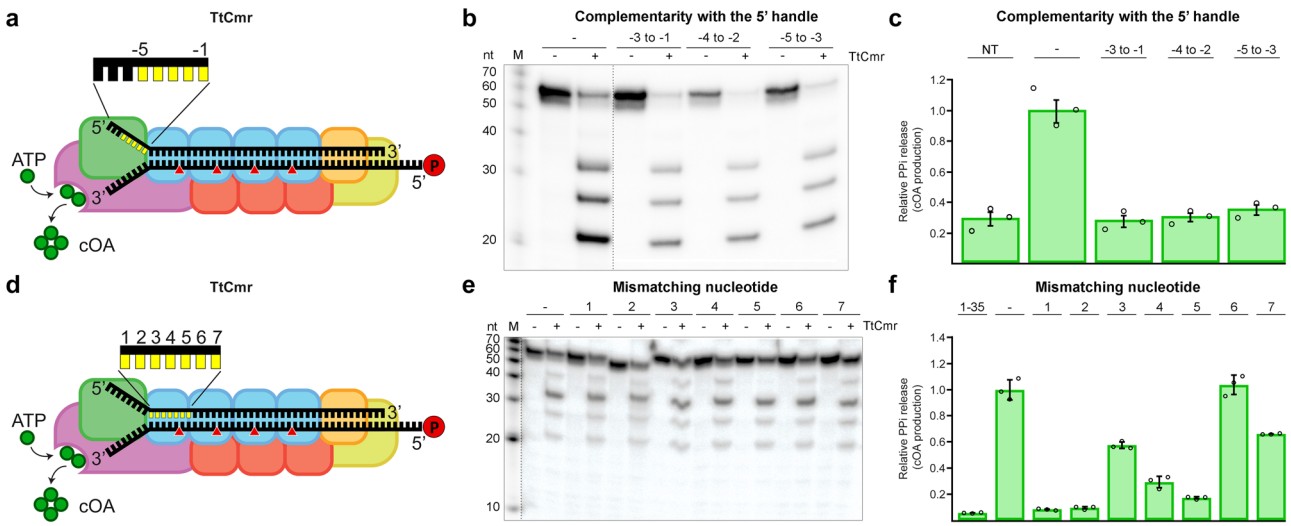

**Fig. 2 Impact of complementarity in the 5′ handle and mismatches in the first spacer region on RNA targeting and cOA production. a** Schematic illustration of the TtCmr complex bound to a target RNA (4.5 target RNA, Supplementary Table S1), showing the different subunits in different colors, crRNA (top strand) and target RNA (bottom strand). The target RNA was labeled at the 5′ end with $^{32}$P ("P" in the red circle). Red triangles indicate the cleavage sites within TtCmr. Highlighted in yellow are the first 5 nucleotides in the 5′ handle of the crRNA (nucleotides −1 to −5). **b** Different target RNAs (Supplementary Table S1) with matches to the 5′ handle were used in an activity assay and analyzed on denaturing PAGE. **c** Impact of 5′ handle complementarity on PPi release as a measure of COA production. **d** Similar schematic illustration as panel **a** with the first 7 nucleotides of the spacer region of the crRNA highlighted in yellow. **e** Similar activity assay as in panel **b** but with RNA targets (Supplementary Table S1) with single mismatches in the first 7 nucleotides of the spacer region of the crRNA. **f** Impact of mismatches in the target RNA with the first 7 nucleotides of the spacer region of the crRNA on PPi release, as a measure for COA production. Discontinuous gel lanes are indicated by a dashed line. The results of the cleavage assays and cOA production assays displayed in this figure are representative results of three replicates (Supplementary Fig. S6). Error bars represent the standard deviation of the mean. Source data are provided as a Source data file.

nt of the spacer region of the crRNA, which is the region on the crRNA base pairing with the protospacer (Fig. 2d). The results showed that RNA targeting was not affected by these mutations, although a mismatch at position 5 abolished cleavage at the adjacent site, as demonstrated by the missing 39 nt degradation product (Fig. 2e). Interestingly however, cOA production was greatly affected by these mismatches, in particular at positions 1 and 2 (Fig. 2f). Similar results were obtained with the reconstituted TtCmr-46 and TtCmr-40 complexes (Supplementary Fig. S2). These results indicate that the seed is either lacking or located in a different region of the crRNA. However, since base pairing at the most 5′ region does appear to be critical for cOA production, we designated this segment as the Cas10-activating region (CAR).

Since the seed is defined as the region on the crRNA that initiates base pairing with its target, we performed EMSA binding assays with the endogenous TtCmr complex (Supplementary Fig. S3). To probe for regions crucial for initiating base paring, we used RNA targets with different mismatching segments (Fig. 3a). We observed that targets with mismatches in the first three segments (nucleotides 1–5, 7–11 and 13–17) did not influence the binding of the target RNA by the TtCmr complex as the migration was similar to that of the fully complementary RNA target control (WT). However, mismatches in the fourth and fifth segments (nucleotides 19–23 and 25–29) substantially affected the electrophoretic mobility of the TtCmr/crRNA-target RNA tertiary complex, suggesting a seed region at the 3′ end of the crRNA.

To further investigate the impact of such a uniquely located seed on RNA degradation and cOA production, we performed activity assays with the endogenous TtCmr complexes using RNA targets containing different mismatching segments (Fig. 3a–c). In agreement with our previous findings, RNA targets with

mismatches in the first segment (S1, nucleotides 1–5) did not interfere with target degradation, despite skipping one cleavage site downstream of the mismatched segment. Similarly, mismatching of the segments S2-S5 (nucleotides 7–12, 13–17, 19–23 and 25–29) resulted in skipping both the adjacent (up- and downstream) cleavage sites, whereas cleavage at the other sites was unaffected. Mismatches in segment S6 (nucleotides 31–35) had no effect on RNA degradation, other than skipping the upstream cleavage site (Fig. 3b). In contrast, some of the mismatching segments substantially affected the cOA production (Fig. 3c). In agreement with results in Fig. 2f, segment mismatches in the CAR (S1 and S2 region) completely abolished the production of cOA. Mismatches in segments S3, S5 and S6 had a minor impact on the production of cOA, whereas a major effect on cOA production was observed with mismatches in segment S4.

Since the endogenous complex is a mixture of longer and shorter complexes, we switched to using the TtCmr-46 or TtCmr-40 reconstituted complexes in order to pinpoint this crucial region more precisely (Fig. 3d–i). The TtCmr-46 complex almost completely mirrored the results obtained with the endogenous complex, with the exception that mismatches in segment S4 seem to abolish the RNA-targeting activity (Fig. 3e). Similarly, mismatches in the CAR (S1 and S2) as well as in S4 diminish cOA production. However, this essential region appeared to have shifted one segment in the TtCmr-40 complex, with strict base pairing requirements for RNA targeting in the third and fourth segments (Fig. 3h). Again, effects on cOA production mirrored these results (Fig. 3i). Taken together, these results demonstrate the existence of a 3′ located seed region in TtCmr that shifts towards the 5′ end of the crRNA in case of smaller guides (in smaller TtCmr complexes). We propose that together, these regions act as flexible seed sequences in TtCmr.

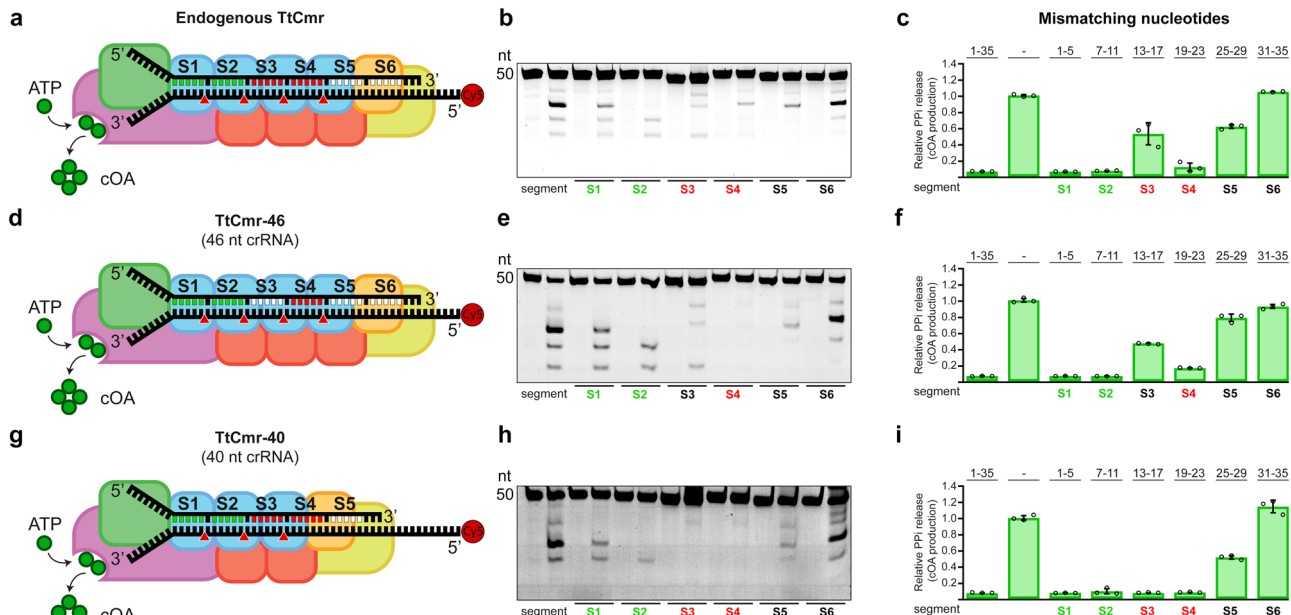

**Fig. 3 A flexible seed region at the 3′ end of the crRNA. a** Schematic illustration of the endogenous TtCmr complex. **b** Different target RNAs with segments mismatches were used in an activity assay and analyzed on denaturing PAGE. **c** Impact of target RNAs with mismatches in the indicated segments on the production of cOA. **d** Schematic overview of the 46 nt crRNA complex (TtCmr-46). **e** Similar to panel **b**, using the 46 nt crRNA complex (TtCmr-46). **f** Similar to panel **c**, using 46 nt crRNA complex (TtCmr-46). **g** Schematic overview of the 40 nt crRNA complex (TtCmr-40). **h** Similar to panel **b**, using the 40 nt crRNA complex (TtCmr-40). **i** Similar to panel **c**, using the 40 nt crRNA complex (TtCmr-40). Target RNA contain a 5′ end Cy5 label (red circle); mismatched segments are indicated with S1–S6. Red triangles indicate the cleavage sites within TtCmr. CAR segments are indicated in green, seed segments are indicated in red. The results of the cleavage assays and cOA production assays displayed in this figure are representative results of three independent experiments (Supplementary Fig. S6). Error bars represent the standard deviation of the mean. Source data are provided as a Source data file.

**Seed of TtCmr structure allows for structural rearrangements.** To determine the structural basis for the 3′ seed in RNA targeting, we interrogated our previously determined cryo-electron microscopy structures of TtCmr with both a 46 and 40-nt crRNA (EMD-2898 and EMD-2899, respectively)[23]. In both structures, the 3′ end of the crRNA is largely exposed, as it is only cradled by the Cmr1/6 heterodimer along one side of the RNA strand (Fig. 4a, b). In contrast, the seed region of the crRNA immediately upstream of the 3′ end (23–38 nt) is partially buried and sandwiched between the Cmr4 and Cmr5 subunits and are less exposed (Fig. 4c, d). The 3′ end of the crRNA is thus primed for transmitting conformational changes and repositioning Cmr5 subunits along the complex to facilitate complete target binding. This strongly suggests that complementarity between the crRNA and target in this region is critical for propagation of base-pairing along the length of the complex, and that it is sensitive to mismatches. Importantly, because the crRNA is shortened by one segment (6 nucleotides) in TtCmr40 compared to TtCmr46, the seed shifts towards the 5′ end of the crRNA (17–32 nt) in the smaller complex. Interestingly, recent structures of the *S. islandicus* Cmr complex also reveal an exposed 3′ end of the crRNA[41]. This indicates that the exposed 3′ end is a common feature among Cmr complexes from different organisms.

**SCOPE - a TtCmr-based nucleic acid detection tool.** Based on these stringent target RNA requirements (i.e., the need for high target complementarity at the seed and the CAR to initiate cOA production), we concluded that type III CRISPR-Cas systems have a high potential for being repurposed as a novel, highly sensitive, robust nucleic acid detection tool. To investigate this possibility, we opted to couple the production of the second messenger to an easy read-out. In nature, these cOAs specifically bind to proteins with a CARF domain, causing an allosteric activation of fused enzyme domains. A well-characterized example of such an CARF-associated enzyme is a cOA-dependent nonspecific RNase (TTHB144) of *T. thermophilus* HB8[30]. We selected this enzyme to establish a synthetic signal transduction route consisting of an RNA-targeting TtCmr/crRNA complex that generates cOA molecules, which in turn trigger the cleavage of a reporter RNA by TTHB144 thereby generating a detectable fluorescence signal.

We first performed in vitro activity assays, using the TtCmr-46 complex, to which we added purified TTHB144 and a 5′ Cy5-labeled reporter RNA. We observed defined degradation products of the reporter RNA only when both the 4.5 target RNA (T) and TTHB144 are present (Fig. 5a), whereas a non-target RNA (NT) did not induce this activity. A guide/target mismatch at the seed region (segment 4 (S4) of TtCmr-46) greatly diminished the intensity of reporter RNA degradation products. In agreement with earlier results, guide/target mismatches in the CAR (1–5 mismatches in segment 1) completely abolished TTHB144 activation, as seen by the lack of reporter RNA degradation products (Fig. 5a). These results show that the target RNA sequence requirements to activate TTHB144 perfectly match with those of cOA production, and that our setup can discriminate single nucleotide differences.

To generate an easy read-out for our tool, we performed a similar assay, with a fluorophore-quencher reporter RNA and measured fluorescence in real-time. For this assay, we used a crRNA that targets the E-gene of the SARS-CoV-2, one of the genes used in various RT-qPCR tests for the corona virus, validated by FIND[42] (Supplementary Table S1). Using this setup, a minimal detectable concentration of 1 nM target RNA concentration was achieved with a fluorescence signal detectable within seconds after starting the incubation (Fig. 5b).

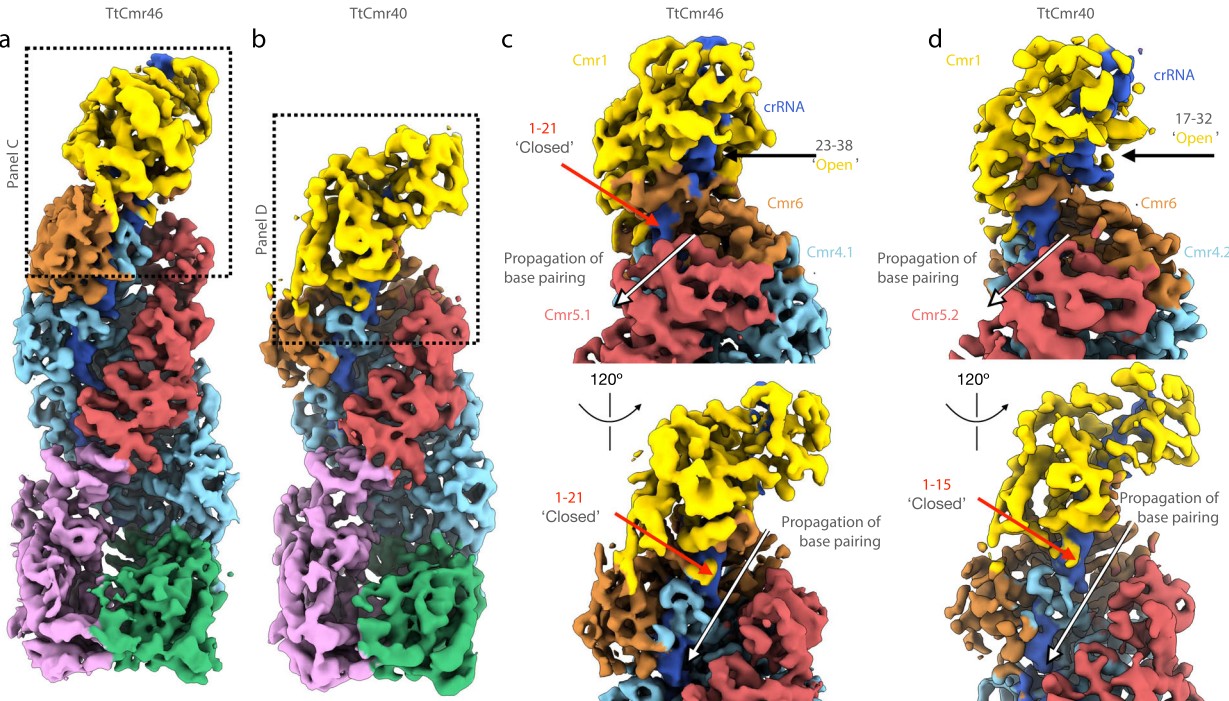

**Fig. 4 Structural basis for flexible 3′ seed region in TtCmr complexes. a** Overall structure of TtCmr complex with 46 nt crRNA (EMD-2898). **b** Overall structure of TtCmr complex with a 40 nt crRNA (EMD-2899). Complexes are colored as in Fig. 1. Boxed regions refer to close-up views shown in panels **c** and **d**. **c** Close-up view of the top of TtCmr46. **d** Close-up view of the top of TtCmr34. The 'open' seed region located at the 3′ end of the crRNA is more accessible than the upstream regions located towards the 5′ end, which are protected by Cmr5 subunits. The 3′ end of the crRNA is thus primed for propagating conformational changes and repositioning Cmr5 subunits along the complex upon target binding.

To enable DNA detection and to enhance the sensitivity of our tool even further, we included a pre-amplification step akin to those used by other CRISPR-Cas based diagnostic tools (Cas12a, Cas13)[43]. For this proof of principle, we designed a (RT-)LAMP pre-amplification step that specifically amplifies the SARS-CoV-2 E-gene, simultaneously adding a T7 promotor to the amplicon that allowed for a subsequent in vitro transcription step[44] (Supplementary Table S1, Fig. 5c). To determine the limit of detection (LOD) of this 2-step approach, ten-fold dilutions of the complete synthetic RNA genome of SARS-CoV-2 were used. The inclusion of the LAMP pre-amplification indeed enhanced the sensitivity of our tool, reaching sensitivities in the atto-molar ($10^{-18}$ M) range within a timespan of ~35 min (30 min pre-amplification (1) + 5 min CRISPR detection (2)) (Fig. 5d).

To validate our 2-step setup in a more complex reaction environment, we continued by testing human nasal swab samples, which were collected at an on-site SARS-CoV-2 testing facility. Out of the 80 samples tested, our tool scored 62 of them positively. We validated these results by comparing them to a PCR-based test (current standard for SARS-CoV-2 testing) that was performed on the same samples in parallel, which were in excellent agreement up to a relevant Ct-value of ~37. In addition, 20 samples that were considered negative by PCR (Ct value not determined, ND), were scored negative by SCOPE as well (Fig. 5e, Supplementary Table S2 and Supplementary Fig. S5).

Lastly, the thermophilic nature of the type III proteins (Thermus-derived Cmr complex and CARF-RNase) of our tool offers an attractive opportunity for a one-pot reaction by combining the LAMP pre-amplification step with CRISPR detection, as the optimum temperature of the reactions is in the same range. Due to the maximum temperature tolerance of an appropriate, commercially available RNA polymerase (Hi-T7 RNAP, NEB), we performed a one-pot LAMP-CRISPR assay at a temperature of 55 °C. The limit of detection was determined at

800 aM, using a synthetic version of the SARS-CoV-2 E-gene, demonstrating the feasibility of this approach (Fig. 5f).

## Discussion

Recent advancements in our understanding of type III CRISPR-Cas systems have highlighted that they have unique mechanistic features compared to other CRISPR-Cas systems. Examples of this include the requirement for reverse-transcriptase activity for some type III systems during the adaptation phase[45–47] and the potentially large signaling network mediated by cOA molecules in the interference phase[27–29,48–50]. Furthermore, type III systems are exceptional in the sense that they are the only CRISPR-Cas system characterized to date capable of targeting both RNA (guide dependent) and DNA (collateral). However, the latest classification in type III CRISPR-Cas systems suggested that not all type III systems are endowed with DNase activity, due to an inactivated or missing HD domain in Cas10[51]. This suggests that RNA is the bona fide target of these systems, as is the case for the type III-B system (Cmr-β) of *S. islandicus*[11], and of *T. thermophilus* presented here.

Yet another unique feature of type III systems is the variable crRNA length with a typical 6 nt periodicity, which, in turn, corresponds to the variable number of Cas7 subunits that constitute the backbone of type III-A and type III-B complexes[8,12,18,20,34,52,53]. Consequently, the cellular population of type III complexes are a heterogenous mixture of complexes with at least three different sizes[12,32,34]. The biological significance of these observations has long remained elusive (see below). Lastly, studies addressing the seed in type III systems are somewhat conflicting, with one report proposing a complete absence of a seed[54], whereas other studies report a seed region in either the 5′[40,55] or 3′ end[56,57] of the crRNA guide. These discrepancies can be explained by the different methods used to

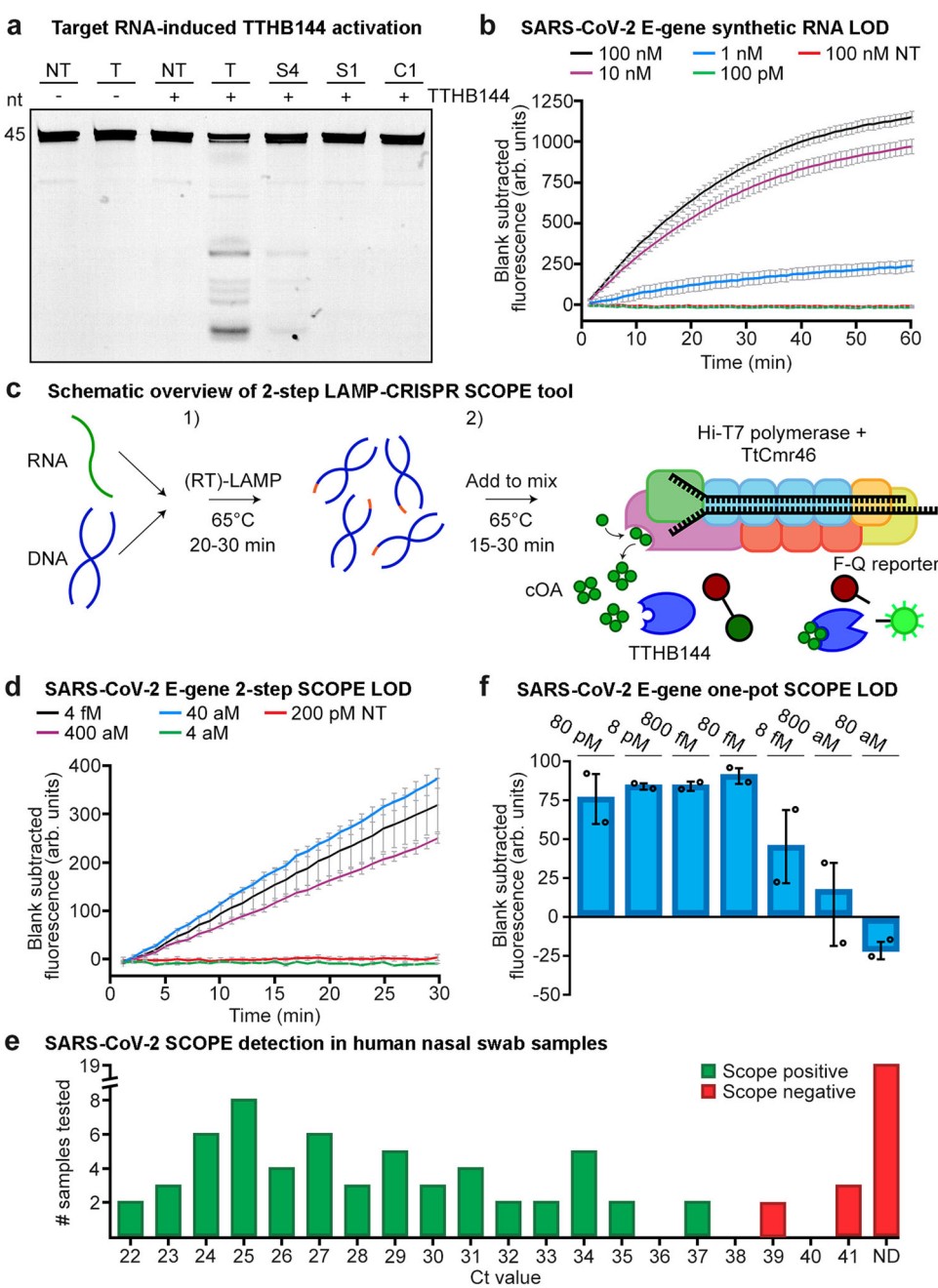

**Fig. 5 A novel type III CRISPR-Cas tool for the sensitive detection of nucleic acids. a** Denaturing PAGE resulting from activity assays performed with the CARF protein TTHB144 and a 5′ Cy5-labeled reporter RNA. Activation of TTHB144 due to cOAs produced by the reconstituted TtCmr-46 complex was monitored by offering fully complementary (T) target RNAs, or target RNA with mismatches in segments one (S1), four (S4) or by a single mismatch in the CAR (C1). A fully non-target RNA (NT) was used as a control. Results of the TTHB144 cleavage assay is representative of results obtained from three replicates (Supplementary Fig. S6). **b** Limit of detection (LOD) assay using reconstituted Cmr-46 on a synthetic SARS-CoV-2 E-gene and a fluorophore-quencher reporter RNA masking construct measured over time. A non-target RNA (NT) was used as a control. Data was obtained from three replicates. **c** Schematic overview of the 2-step reaction setup consisting out of (1) a (RT)-LAMP based pre-amplification step and (2) a T7-based in vitro transcription and type III CRISPR detection step. 'F-Q' represents the fluorophore-quencher reporter RNA. **d** Limit of detection assay using reconstituted Cmr-46 in the 2-step setup (depicted in panel **c**), with a SARS-CoV-2 synthetic full RNA genome as target. Data was obtained from three replicates. **e** Detection of SARS-CoV-2 in human swab samples. Ct-values of qPCR analysis (*Orf1ab* gene) of 81 samples are depicted on the X-axis with the true negative samples displayed as not determined (ND). See Supplementary Fig. S5 and Supplementary Table S2 for Ct-values of qPCR analysis of SARS-CoV-2 samples and respective Scope tool score. The data on the qPCR and SCOPE analysis was obtained from one replicate measurement. **f** One-pot LAMP-CRISPR limit of detection assay, using reconstituted Cmr-46, on a synthetic SARS-CoV-2 E-gene. Data was obtained from two replicates. All error bars in this figure represent the standard deviation of the mean. 'Arb. units' stands for arbitrary units. Source data are provided as a Source data file.

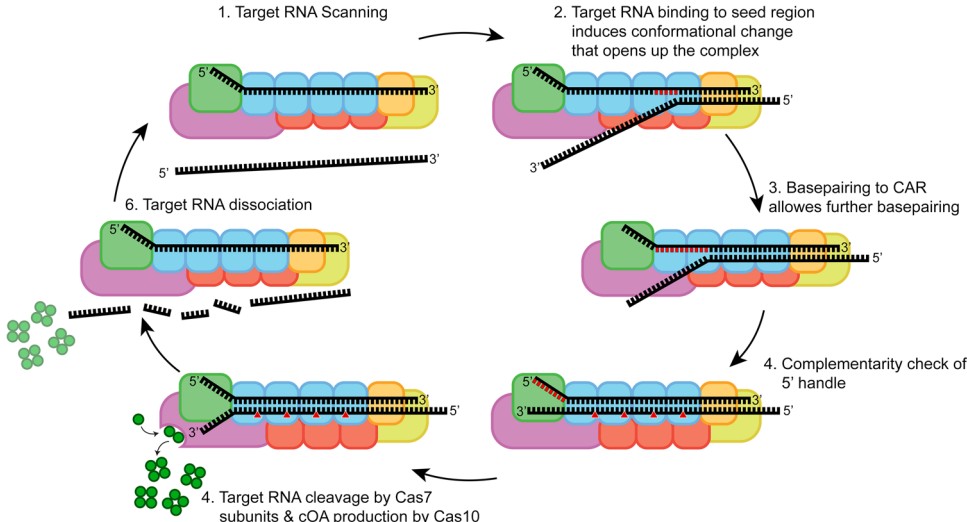

**Fig. 6 Schematic model of TtCmr target binding and subsequent activities. (1)** TtCmr complex with bound crRNA is scanning for complimentary target RNA. **(2)** Target RNA binding is initiated at seed region, which induces a conformational change that allows further base pairing. **(3)** Full base pairing of target RNA to the crRNA, activating Cas10. **(4)** Target RNA is cleaved by Cas7 subunits and cOA is produced by Cas10. **(5)** Cleaved target RNA dissociates from TtCmr.

pinpoint certain regions of importance on the crRNA. For instance, read-outs that either directly or indirectly look at cOA production, such as phage challenge experiments or conformational changes in Cas10, will point towards a bigger importance of the 5′ region. However, studies on the relationship of seed requirements and cOA production have paved the way for a more thorough and complete investigation[49]. Here, we dissected the importance of different size-variants of the TtCmr complex and the different regions on the crRNA by looking at each step from target RNA binding to CARF protein activation individually. Together, our data resulted in a model presented in Fig. 6.

Firstly, we demonstrated that target RNA binding is initiated by a largely exposed 3′ region on the crRNA, which we propose should be designated as the seed region (Fig. 3 and Supplementary Fig. S3). This region coincides with the first region of the crRNA leaving the Cas7/Cas11 backbone that extends to Cmr1 and Cmr6 on that side of the complex. In agreement with our observations, previous studies showed that target RNA binding in the type III-B system requires Cmr1 and Cmr6, and that base pairing in a confined region within the 3′ end of the crRNA is crucial for interference[24,53]. We therefore propose that Cmr1 and Cmr6 are involved in the proper positioning of the seed region to initiate base pairing with its cognate target RNA. The position of Cmr1 and Cmr6 on the crRNA is determined by the variable 3′ end of the crRNA. Indeed, we observed that shorter complexes (with the 40 nt crRNA) had a seed region, that was shifted one segment towards the 5′ end (Fig. 3h–i). We conclude that target RNA binding (and cleavage) is governed by a flexible 3′ located seed region. This is an important difference with the structurally-related type I complexes, which harbor a fixed 5′ seed region[37].

Secondly, base pairing at the seed region promotes a conformational change within the TtCmr complex, opening a channel along the Cas7/Cas11 backbone wide enough to accommodate further base pairing interactions of the crRNA with its target RNA (Figs. 4 and 6)[23]. Our results show that, in contrast to target RNA degradation, cOA production is dependent on base pairing with the first seven 5′ nucleotides of the spacer part of the crRNA (with the exception of the nucleotide at position six), a region that we designated as the CAR (Cas10-Activating Region). In agreement with previous work on type I and type III complexes, the nucleotide on the crRNA after each segment (i.e., every

sixth nucleotide) is excluded from base pairing with the target RNA, due to the thumb-like extension of Cas7[23,37].

Lastly, even after fulfilling RNA-targeting requirements (starting at the seed, followed by the CAR) we showed that a final checkpoint ensures that no cOA is produced when targeting self-RNAs (i.e., antisense transcripts from the CRISPR array, which has been reported to occur occasionally)[58]. Although self-RNAs are bound and cleaved by TtCmr, the complementarity between the 5′ handle and the corresponding region on the self-RNA does completely prevent cOA production.

Although the biological significance of our findings still awaits further investigation in vivo, we anticipate that the flexibility of the seed region in type III systems will lower the chances that MGEs escape CRISPR-Cas targeting by introducing mutations in the protospacer. In type I CRISPR-Cas systems, MGE escapees promote the rapid acquisition of additional spacers, in a process called primed adaption[59,60], which ensure within-host spacer diversity. Our findings suggest that type III systems can create this diversity with just a single spacer. On one hand, the ability to recognize heavily mutated MGEs might be beneficial for the host to provide a robust interference response towards MGEs[54,61]. In *Marinomonas mediterranea* for example, a horizontally acquired type III-B system was recently shown to effectively bolster the immune response of its native type I-F systems to cope with phage escapees[62]. On the other hand, this same flexibility might come at the cost of a higher risk of self-targeting. While incidental binding and degradation of self-RNAs by itself might not represent a large fitness cost to the host, the subsequent activation of the Cas10 Palm domain (resulting in the production of the second messenger that allosterically activates for example promiscuous, sequence nonspecific RNase activity) might have more detrimental consequences[15,16]. This scenario is in good agreement with our results, showing that activation of the Cas10 Palm domain is indeed only induced when very specific conditions are met (matching seed, matching CAR and non-matching 5′ handle).

The stringent control of cOA production has motivated us to repurpose type III CRISPR-Cas systems for the specific detection of nucleic acids. We first developed a novel pyrophosphatase-based colorimetric assay, which allows for easy quantification of oligoadenylate production. When combined with appropriate Pi

calibration curves, this method could further be developed for the absolute quantification cOAs or even to quantify the virus titers in a sample. For signal amplification purposes and to achieve an easy read-out (RNase activity), we selected TTHB144 - one of the three native cOA-activatable (CARF) proteins present in the genome of *T. thermophilus*[27,30,63]. While TTHB152 is also characterized as a CARF-RNase, it is encoded in the type III-A operon, so for consistency we opted to use TTHB144. We observed highly similar trends in target RNA sequence requirements (CAR and segments mutants) for activation of the RNase activity of TTHB144 compared to those governing cOA production (Figs. 2 and 3). This shows that the stringent control of TTHB144 activation could be utilized to make the highest possible distinctions between target RNAs, i.e., monitoring even a single nucleotide difference (Fig. 5a, C1 mutant). Next to the specificity aspect, due to the combination of an intrinsic signal amplification step (Cas10 producing a multitude of cOA molecules) and the high ribonuclease activity of TTHB144[30], our tool was able to very quickly generate a fluorescence signal: detecting 1 nM of target RNA within seconds (Fig. 5b). For most diagnostic applications however, a sensitivity of 1 nM is not high enough[43]. Therefore, we adapted a previously established SARS-CoV-2 LAMP pre-amplification reaction to boost the sensitivity of our diagnostic tool even further[44]. Indeed, after testing this 2-step protocol (Fig. 5c) on a SARS-CoV-2 RNA reference genome, we determined its limit of detection at 40 aM (~25 copies/µL). Validating our test on human nasal swab samples indicated our test to be 100% accurate up to a Ct value of ~37, which is higher than most accepted cut-offs for diagnosis[42]. Equally important, we did not detect any false positives out of the 20 negative samples tested.

Ideally, a true one-pot reaction is preferred to minimize steps and to reduce the risk of cross-contamination. We therefore investigated the performance of our system in a one-pot setup, at a temperature of 55 °C (to comply with the upper temperature limit of Hi-T7 polymerase). Despite using sub-optimal temperature for the CRISPR detection step, we were able to achieve a limit of detection of 800 aM using a synthetic gene template (Fig. 5f). We showed the feasibility of the one-pot LAMP CRISPR detection approach with some room for optimization regarding the amount of signal generated and reducing the incubation time. Although similar reaction component concentrations compared to the 2-step protocol were used, both the LOD (800 aM) and reaction time (180 min) were affected in the one-pot approach. This is likely due to the sub-optimal incubation temperature of 55 °C combined with potential interfering reaction components to the LAMP reaction. Optimization consisting of changing TTHB144 and reporter RNA concentrations, for example, could improve this issue. Furthermore, the LAMP reaction by itself could be designed and optimized to work more efficiently at a temperature of 55 °C by choosing a different DNA polymerase or change the primer design to reduce overall reaction times.

A couple of CRISPR-based nucleic acids detection platforms have been developed over the last years, such as the Class2-based DETECTR and SHERLOCK platforms[64,65]. However, SCOPE is the first Class 1-based CRISPR-Cas nucleic acid detection tool, with some very useful characteristics: highly sensitive (requiring very little sample input, see methods) and specific (flexible seed, stringent CAR), quick (detection within seconds), flexible (PAM-independent guide design), highly robust (long shelf-life). Furthermore, the potential for a 1-pot system in combination with RT-LAMP offers opportunities to increase the throughput of our tool. The thermophilic nature of the system potentially means that the system is less affected by, enzyme mediated, inhibitory factors in crudely extracted samples. While the 2-step approach currently limits efficient high-throughput testing due to the extra

(manual) step that is required, an efficient 1-pot system would mitigate this. The use of a standard fluorophore and no proprietary plastics further facilitates integration in current high throughput testing facilities. All these favorable features make the SCOPE approach a highly attractive alternative over the currently available detection tools.

## Methods

**Purification of the Cmr complex and individual subunits.** Harvested *T. thermophilus* HB8 cells, producing the (His)$_6$-tagged Cmr complex, were resuspended in 100 ml of 50 mM NaCL, 20 mM Tris-HCl (pH 8.0). Subsequently, these cells were lysed by sonication in ice water, and spun down at 200,000 × g for 1 h at 4 °C. The supernatant was used for purification using a HisTrap HP column (GE Healthcare), pre-equilibrated with 150 mM NaCl, 20 mM Tris-HCl, 20 mM imidazole (pH 8.0), after which elution was done using a linear gradient of 20–500 mM imidazole. Collected fractions were desalted using a HiPrep 26/10 desalting column (GE Healthcare). The sample was then applied to a RESOURCE Q column (GE Healthcare), pre-equilibrated with 20 mM Tris-HCl (pH 8.0), and the bound proteins were eluted with a linear gradient of 0–0.5 M NaCl. Finally, gel filtration was performed using a HiLoad 16/60 Superdex 200 pg (GE Healthcare) column, pre-equilibrated with 150 mM NaCl, 20 mM Tris-HCl (pH 8.0). The collected fractions were concentrated with a Vivaspin 20 concentrator (30,000 Da molecular weight cut-off, Sartorius)[18]. For reconstituted complex, each subunit was individually expressed and purified. All subunits were cloned in a bicistronic design elements containing expression plasmid under an Isopropyl β-D-1-thiogalacto-pyranoside (IPTG) inducible T7 promoter and N-terminal Streptavidin tag. Expression plasmid containing *E. coli* BL21(DE3) were grown at 37 °C until ~OD$_{600}$ = 0.6, after which the culture was cold-shocked on ice for 1 h. IPTG was added to a final concentration of 0.5–1 mM and the culture was incubated at 20 °C for 16 h. Cells were harvested and resuspended in Wash Buffer (150 mM NaCl, 100 mM Tris-HCl, pH 8) and a Complete protease inhibitor tablet was added (Roche). Cells were lysed by sonication (25% amplitude 1 sec on, 2 sec off, Bandelin Sonopuls) and spun down at 30.000 × g for 45 min, subsequent lysate was filter (0.45 µm) clarified. A StrepTrap HP (GE) column was equilibrated using Wash Buffer and the lysate was run over it. Wash Buffer with d-Desthiobiotin added to a final concentration of 2.5 mM was used to elute. The elution fractions were pooled, concentrated and run over a HiLoad® 16/600 Superdex® 75 pg size exclusion chromatography column for further purification. Purification of TTHB144 was performed in similar fashion.

**Reconstitution of TtCmr complexes.** For TtCmr46, 3.5 µL crRNA (700 ng) was added to 3.5 µL 1X Cmr buffer (20 mM Tris-HCl pH 8.0, 150 mM NaCl). Subsequently, the subunits were added to the reaction mixture in a specific order (Cmr3, Cmr2, Cmr4, Cmr5, Cmr6, Cmr1) to a final concentration of 2.5 µM, 2.5 µM, 10 µM, 7.5 µM, 2.5 µM and 2.5 µM respectively, to make up a total reaction volume of 20 µL. For TtCmr40 final concentration of subunits was adjusted to 2.5 µM, 2.5 µM, 7.5 µM, 5 µM, 2.5 µM and 2.5 µM respectively. The reaction mixture was incubated at 65°C for 30 min.

**In vitro cleavage activity assays.** RNA substrates (listed in Supplementary Table S1) were either 5′ labeled by T4 polynucleotide kinase (NEB) and 5′ $^{32}$P-γ-ATP, after which they were purified from a denaturing PAGE using RNA gel elution buffer (0.5 M Sodium acetate, 10 mM MgCl$_2$, 1 mM EDTA and 0.1% SDS) or ordered with a 5′ Cy5 fluorescent label. In vitro cleavage activity assays were conducted in TtCmr activity assay buffer (20 mM Tris-HCl pH 8.0, 150 mM NaCl, 10 mM DTT, 1 mM ATP, and 2 mM MgCl$_2$) using the RNA substrate and 62.5 nM TtCmr. Unless stated otherwise, the reaction was incubated at 65 °C for 1 h. RNA loading dye (containing 95% formamide, dyes left out in case of Cy5 substrates) was added to the samples after incubation and boiled at 95° for 5 min. The samples were run on a 20% denaturing polyacrylamide gel (containing 7 M urea) for about 1–4 h at 15 mA or overnight at a constant of 4 mA. The image was visualized using phosphorimaging or fluorescent gel scanning (GE Amersham Typhoon).

**cOA detection assay.** The in vitro cOA detection assays were conducted in TtCmr activity assay buffer (20 mM Tris-HCl pH 8.0, 150 mM NaCl, 10 mM DTT, 1 mM ATP, and 1 mM MgCl$_2$) to which Cmr-complex (62.5 nM final concentration) as well as the RNA substrates (200 nM, listed in Supplementary Table S1) were added. The reaction was incubated at 65 °C for 1 h after which 0.05 units of pyrophosphatase (ThermoFisher EF0221) were added, followed by an incubation at 25 °C for 30 min (Fig. 2). Alternatively, thermostable pyrophosphatase (NEB #M0296) was added during the 1 h incubation at 65 °C (Fig. 3). cOA quantification was achieved by using the Malachite Green Phosphate Assay Kit (Sigma-Aldrich MAK307). This resulted in an OD$_{650}$ signal, which was measured on a BioTek Synergy Mx platereader using the BioTek Gen5 software.

The unitless relative PPi levels (cOA-production) presented in Figs. 2 and 3 were calculated by expressing the OD$_{650}$ signal from the individual RNA targets (containing the indicated mismatches) as a ratio of the OD$_{650}$ signal obtained from the fully complementary target RNA.

**EMSA**. EMSAs were performed by incubating 62.5 nM TtCmr complex with 13.3 nM 5′ Cy5-labeled target RNAs (Supplementary Table S1) in Cmr binding buffer (20 mM Tris-HCl pH 8.0, 150 mM NaCl, 0.1 mM DTT, 1 mM EDTA). All reactions were incubated for 20 min at 65 °C before electrophoresis on a native 5% (w/v) polyacrylamide gel (PAGE), running at 15 mA. The image was visualized via fluorescent gel scanning (GE Amersham Typhoon).

**Structural modeling**. In order to model crRNA within our previously determined TtCmr46 and TtCmr40 maps (EMD-2898 and -2899, respectively), the model of *S. islandicus* Cmr complex (PD 6S6B) was fitted as a single rigid body into the TtCmr46 map, and supervised flexible fitting was performed using Isolde[66]. Maps were visualized using ChimeraX[67].

**Human swab sample collection, extraction and qPCR**. Combined human throat/nasopharyngeal swab samples were obtained at a community testing center (Utrecht, the Netherlands) from adults with either SARS-CoV2- related symptoms or with contacts of infected persons. Recruitment occurred at a testing site that was set up as a research site to evaluate new diagnostic tests for SARS-CoV-2. Adults who visited the testing center were asked to participate in validation studies of novel diagnostic tests. Participants gave consent to use of residual material of the obtained combined human throat/nasopharyngeal swab samples. As samples were collected within the framework of the national COVID measures and testing efforts, samples were routinely collected at interval timepoints. The medical research ethics committee (MREC) of Utrecht decided that validation studies of new diagnostic tests for SARS-CoV-2 is not subject to the Medical Research Involving Human Subjects Act (WMO) and did not require full review by an accredited MREC. All participants were informed and consented with participation. PCR was conducted in a certified clinical laboratory and all procedures were validated according to the ISO 15189 standard. Nasopharyngeal swabs were transferred into 3 ml Universal transport medium. RNA was isolated and purified using the MagC extraction kit (Seegene) on an automatic nucleic acid extractor Hamilton MicroLAB StartLET (Bonaduz). SARS-CoV-2 qPCR was performed using the HKU protocol for ORF1ab gene detection. Primers and probe (Supplementary Table S1) were used in combination with TaqMan Fast Virus Master mix (ThermoFisher # 4444432) and incubated according to the following protocol: (1x) [50 °C, 5 min | 95 °C 20 sec], (40x) [95 °C, 5 sec | 60 °C 30 sec][68].

Results were interpreted with the 7500 Fast SDS (Applied Biosystems) data analysis software. A positive result was defined as amplification of the *Orf1ab* SARS-CoV-2 gene.

**Nucleic acid detection tool**. The target RNA induced activation of TTHB144 assays were performed similarly to the earlier described in vitro cleavage assays using reconstituted TtCmr-46 complex. However, non-labeled (non)target RNA was used and either a 5′ Cy5 reporter RNA (Fig. 5a) or commercial RNaseAlert (Fig. 5b, d, f and e) was added, as well as 1 μM purified TTHB144.

The LAMP reaction in the 2-step LAMP-CRISPR detection setup (Fig. 5c, d) was performed using the WarmStart® LAMP Kit (NEB #E1700), using previously published primers concentration described in literature (Supplementary Table S1), designed for amplifying SARS-CoV-2 RNA[44]. Final primer concentrations of 0.2 μM, 1.6 μM and 0.8 μM for the F3/B3, FIP/BIP and LoopF/LoopB primers were used respectively. A T7 promotor sequence was added to the loop primer to allow for the subsequent in vitro transcription reaction. After pre-amplification by LAMP, subsequent T7 polymerase transcription will enrich the target RNA, partly consisting of a template derived region which lies between the Loop primer and the B2 primer (Supplementary Fig. S4). The 2-step assay was performed by a 30 min LAMP pre-amplification step, using 5 μL of sample RNA extract, and a 15 min CRISPR detection step. The CRISPR detection step was performed by adding 8.75 μL CRISPR Mix (final concentrations after addition: 1X TtCmr Activity Assay Buffer, ~62.5 nM reconstituted TtCmr-46 complex, 500 nM TTHB144, 2 μL NTP Buffer Mix (NEB #E2050), 25 U Hi-T7 RNA polymerase (NEB #M0658), 250 nM RNaseAlert™ QC System v2 (ThermoFisher) and 8 mM MgCl). Measurements were made at 1- or 2-min intervals at 65 °C on a Bio-Rad CFX96 qPCR machine using the CFX Maestro software (Fig. 5d) and Applied Biosystems™ 7500 using the 7500 Fast SDS software (Fig. 5e) (FAM channel). The SARS-CoV-2 synthetic genome was ordered from Twist Biosciences.

The one-pot LAMP-CRISPR reaction was set up as a total reaction volume of 20 μL, according to the LAMP method described earlier with additions to the earlier described CRISPR Mix (Modification: 1 μL NTP Buffer Mix (From NEB #E2050, no additional MgCl). Total incubation-time for the one-pot reaction was 180 min.

In all cases, the limit of detection (LOD) was determined by the lowest concentration of target RNA in the reaction mixture in step 1, which led to an significant increase in fluorescence compared to the non-target RNA control. The delta-signal SCOPE value was calculated by taking the, blank subtracted, increase of fluorescent signal over time during the 15 min of the CRISPR detection step in the 2-step protocol. Similarly, the same calculation was used for the one-pot assay, by using the signal increase over 180 min of incubation. For the human swap samples presented in Fig. 5e, a sample was considered positive if the average delta-SCOPE value was within 2 standard deviations of all qPCR-scored positive samples.

**Reporting summary**. Further information on research design is available in the Nature Research Reporting Summary linked to this article.

## Data availability

The structures of the different TtCmr complexes (Fig. 4) have previously been deposited into the Electron Microscopy Data Bank (EMDB) under the accession codes EMD-2898 and EMD-2899[23]. Source data are provided with this paper.

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

## Acknowledgements

We thank the Nogales and Doudna labs for their support on the structural analyses (University of California, Berkeley, CA, USA). R.H.J.S. was supported by a VENI grant (016.Veni.171.047), J.V.D.O. by a TOP grant (714.015.001), and S.J.J.B. by a VICI grant (VI.C.182.027), all from The Netherlands Organization for Scientific Research (NWO). This work was supported in part by Welch Foundation grant F-1938 (to D.W.T.), National Institute of General Medical Sciences (NIGMS) of the National Institutes of Health (NIH) R35GM138348 (to D.W.T.) and a Robert J. Kleberg, Jr. and Helen C. Kleberg Foundation Medical Research Award (to D.W.T.). D.W.T. is a CPRIT Scholar supported by the Cancer Prevention and Research Institute of Texas (RR160088) and an Army Young Investigator supported by the Army Research Office (W911NF-19-1-0021). This work was also supported by the David Taylor Excellence Fund in Structural Biology made possible with support from Judy and Henry Sauer (to D.W.T.).

## Author contributions

R.H.J.S., Y.Z., J.v.d.O. and J.A.S. conceived of and designed the study. Y.Z., J.A.S., D.W.T., J.P.K.B., S.H.P.P. and A.S. performed experiments and analyses. R.H.J.S., S.J.J.B., J.v.d.O. provided experimental guidance. R.H.J.S. and J.A.S. wrote the manuscript with significant input from other authors.

## Competing interests

The Authors declare the following competing interests. J.A.S., S.H.P.P. are founders and shareholders of Scope Biosciences. J.v.d.O., R.H.J.S. are shareholders and members of the scientific board of Scope Biosciences. J.A.S., J.v.d.O., R.H.J.S., S.H.P.P. are inventors on type III CRISPR-Cas related patents. Authors Y.Z., D.W.T., J.P.K.B., C.D.S., B.J.F.K., M.O., L.M.H., S.J.J.B. and A.S. declare no competing interests.
