## [Peer Review File · Nature Communications]

Reviewers' Comments:

Reviewer #1:

Remarks to the Author:

This work by Steens et al continues the authors' pioneering work on type III-B CRISPR-Cas system from *T. thermophilus* HB8. In this study, the authors have 1) further clarified the dispute on the seed region for target binding of type III system to be at the 3' end of crRNA, which can shift depending on Cmr stoichiometry; 2) clarified that the target binding of the 5' end of crRNA is required for cOA production, and 3) developed a type III CRISPR-based diagnostic platform called SCOPE. Overall this is an exciting study from mechanistic and tool development standpoints, with detailed work on both fronts, and could become suitable for publication in Nature Communications. However, I find the methods section and detailed descriptions of the experiments to be lacking throughout the manuscript. This is especially problematic with the nucleic acid detection part. The authors should go through their manuscript and edit it extensively, with the goal that readers should be able to replicate their results. Comments which I hope the authors can address in full in their revision are below:

For the biochemical characterization parts:

- For the malachite green assay, what is the calculation the authors did to arrive at the PPI release (CoA production) numbers shown on the y-axes? There is no unit to these values so I presume these are relative values of some sort, but I do not want to presume anything. Please also give detail on how the authors convert the OD650 values from the assay and ensure its linear detection range of phosphate concentrations (i.e. calibration methods).
- There are two variants of the pyrophosphatase enzymes the authors mentioned one can use. Please just specify which variant is used in the manuscript, and for which figure.
- For all the RNA gels the authors showed and also the EMSA result, the authors should specify that these are representative results from a defined number of replicates the authors performed (and give the number of replicates performed).
- For key RNA and EMSA gels, it would be nice to see data from additional replicates in the supporting information. This will give readers more confidence in interpreting many of the authors' exciting data, including the EMSA gel patterns (which point to S4/S5 being the key RNA region for binding).
- For EMSA assays, could the authors speculate what may give rise to the different shift patterns of the S5 mismatch? It does not seem to be a simple loss of binding.
- For EMSA assays, please give the concentration of target RNAs used.
- Figure 2F: could the authors discuss why the mismatch at position 6 results in no change in cOA production? Is there an explanation based on the Cmr complex structure? Is it possible the cOA production pattern upon having mismatches at nucleotide 3-5 could recur at downstream positions?

For the nucleic acid detection part:

- Please do not refer to methods described in earlier literature (e.g. previously performed LAMP protocols); please just inform readers exactly what was done for the experiments shown in this manuscript.
- Can the authors provide more rationale why among CARF-RNases TTHB144 is selected, beyond it being well-characterized?
- Do the authors use endogenous or reconstituted TtCmr for these assays? It seems the reconstituted TtCmr-46 is used for Figure 5A, but unclear about the rest of Figure 5.
- For Figure 5A, please give sequence of the non-target RNA (NT) used as a control.
- Figure 5A, B, D, F: I presume the cy5 reporter is used in Figure 5A, while RNase alert is used in 5B, D, and F. But better to write these details out and leave no ambiguity.
- For the clinical sample assessment, what is the exact reaction time for the LAMP step? What is the exact reaction time for the CRISPR step? Giving time ranges as in Figure 5C is ok as an overview, but insufficient when trying to associate methods with results.
- What was the volume input of the clinical RNA extracts the authors used in their reaction?
- My understanding is the authors prepare the CRISPR mix and add this directly to the LAMP reaction post-amplification. Is this correct? Is the concentration of CRISPR mix components final concentration, after adding to the LAMP reaction?
- What is the delta-signal SCOPE value shown in Table S2? Are these the blank-subtracted

fluorescence similar to what is shown in Figure 5? But why are the numbers from Table S2 and Figure 5 not in the same range?

- Text associated with Figure 5E: the agreement with RT-PCR is up to Ct 37 based on the data, not to Ct 39 as written in the text

- Figure 5F: for the one-pot SCOPE result, what is the SCOPE reaction time that gives rise to this result? In addition to poorer LoD, if the fluorescence signal from the one-pot version is consistently lower than the two-put version, the authors should comment so.

- Figure 5A: there is a mentioning of testing a mismatched sequence in segment 6 for TTHB144 activation in the figure legend, but the result is absent

Reviewer #2:

Remarks to the Author:

The authors describe the development of proof-of-concept diagnostic based on a type III CRISPR-cas platform, SCOPE. A majority of the manuscript focuses on the study of type III CRISPR Cas RNA binding and cleavage mechanisms, and data on diagnostic application is presented for the detection of SARS-CoV-2. Authors report a class 1-based CRISPR-cas nucleic acid detection tool, claiming high specificity and rapid detection. Authors demonstrate fluorescence-based SARS-CoV-2 detection in buffer, detecting 1 nM of target RNA after a few seconds of incubation. In addition, a 2-step protocol using (RT-) LAMP preamplification is described and authors report attomolar limits of detection. This 2-step protocol was tested in 80 human nasal swabs in correlation with PCR measurements. Using a CT threshold of 39, 60 samples were positive by PCR and all 60 were also positive by the SCOPE test. Lastly, authors demonstrate a one-pot synthesis protocol for SARS-CoV-2 detection, reporting a limit of detection of 800 aM.

The manuscript is well-written, with most of the work being an advance that can be special interest to the CRISPR community. Data presented for SCOPE as a diagnostic is intriguing and limits of detection are impressive, but evaluation of test sensitivity is insufficient. CRISPR-based SARS-CoV-2 tests are potential tools to combat the pandemic, however, if the following manuscript is to present SCOPE as a diagnostic tool, rigorous evaluation of test performance and details of test methods should be added to strengthen the claim.

Comments:

1. LOD for the fluorescence-based test, the 2-step test, and the one-pot reaction test are stated but there is no description of how LODs were determined. Clearly state how the LOD was calculated. Include description of replicates for measurements.
2. How was SCOPE signal defined? SI shows delta signal SCOPE, how was this calculated?
3. Authors demonstrated specificity via denaturing PAGE and should perform similar control experiments to validate diagnostic tests. For example, include fluorescence signal generated after incubation for other corona viruses such as hCov NL63, hCov OC43, and hCov 229E, showing data for fluorescence signal vs incubation time and signal at the final incubation time.
4. What is the cut off value to determine positive and negative nasal swabs by SCOPE? What was the criteria for determining this cutoff?
5. Provide brief description of where nasal swabs were obtained from, patients? Healthy adults? COVID19 patients? Were replicate measurements in nasal swabs measured?
6. Authors should briefly discuss advantages and limitations of SCOPE for diagnostic applications including over all test time, sample volumes needed, sample throughput, etc.

Rebuttal - Steens et al. - Manuscript NCOMMS-21-07914

We are thankful for the constructive feedback from the Editor and Reviewers. Below, we have addressed all the Editor's and Reviewers' concerns point-by-point, indicated as "Author's reply" in bold.

EDITORS COMMENTS

Thank you again for submitting your manuscript "SCOPE: Flexible targeting and stringent CARF ribonuclease activation enables type III CRISPR-Cas diagnostics" to Nature Communications. We have now received reports from 2 reviewers and, on the basis of their comments, we have decided to invite a revision of your work for further consideration in our journal. Your revision should address all the points raised by our reviewers (see their reports below).

REVIEWER COMMENTS

Reviewer #1 (Remarks to the Author):

This work by Steens et al continues the authors' pioneering work on type III-B CRISPR-Cas system from *T. thermophilus* HB8. In this study, the authors have 1) further clarified the dispute on the seed region for target binding of type III system to be at the 3' end of crRNA, which can shift depending on Cmr stoichiometry; 2) clarified that the target binding of the 5' end of crRNA is required for cOA production, and 3) developed a type III CRISPR-based diagnostic platform called SCOPE. Overall, this is an exciting study from mechanistic and tool development standpoints, with detailed work on both fronts, and could become suitable for publication in Nature Communications. However, I find the methods section and detailed descriptions of the experiments to be lacking throughout the manuscript. This is especially problematic with the nucleic acid detection part. The authors should go through their manuscript and edit it extensively, with the goal that readers should be able to replicate their results. Comments which I hope the authors can address in full in their revision are below:

Author's reply:

We thank the reviewer for the appreciation of our work on the new mechanistic insights and tool development. We have taken the constructive comments regarding the methods section, which were in agreement with comments by reviewer #2, as a starting point to substantially revise the manuscript to improve replicability. The details of these changes are outlined below.

For the biochemical characterization parts:

For the malachite green assay, what is the calculation the authors did to arrive at the PPi release (CoA production) numbers shown on the y-axes? There is no unit to these values so I presume these are relative values of some sort, but I do not want to presume anything. Please also give detail on how the authors convert the OD650 values from the assay and ensure its linear detection range of phosphate concentrations (i.e. calibration methods).

Author's reply:

The reviewer is correct that the numbers on the y-axes represent normalized, relative values as a measure for PPi release (cOA production) and are therefore without units. These relative values were calculated by expressing the OD650 signal from the individual RNA targets (containing the indicated mismatches) as a ratio of the OD650 signal obtained from the fully complementary target RNA. We agree with the reviewer that this was indeed not explained properly. As such, we have now changed the text next to the y-axis to: "relative PPi levels (cOA-production)" and adjusted the legend accordingly. In addition, we have expanded the methods by explaining how these values were calculated.

The relevant section (lines 646-652) in the methods now reads:

“cOA quantification was achieved by using the Malachite Green Phosphate Assay Kit (Sigma-Aldrich MAK307). This resulted in an OD₆₅₀ signal, which was measured on a BioTek Synergy Mx platereader. The unitless relative PPI levels (cOA-production) presented in Figures 2 and 3 were calculated by expressing the OD₆₅₀ signal from the individual RNA targets (containing the indicated mismatches) as a ratio of the OD₆₅₀ signal obtained from the fully complementary target RNA.”

Although we show a clear correlation between cOA production, CARF protein activation and the PPI levels measured in our malachite green assays, we currently do not know whether this correlation is linear. Because of this, we refrained from making quantitative statements (‘twice as much’ etc) about the differences in cOA production signals between different target RNAs. We hope to be able to make an absolute quantification of the cOAs produced by the type III complex in the future, , as at present we cannot rule out that the used type III complex might produce different cOAs (with different number of AMP moieties in the ring, resulting in more PPI release), which might complicate the read-out.

- There are two variants of the pyrophosphatase enzymes the authors mentioned one can use. Please just specify which variant is used in the manuscript, and for which figure.

Author’s reply:

The standard pyrophosphatase (ThermoFisher EF0221) has been used in Figure 2 while the thermostable pyrophosphatase (NEB M0296) was used in Figure 3. This has now been clarified in the methods section at lines 644-646.

- For all the RNA gels the authors showed and also the EMSA result, the authors should specify that these are representative results from a defined number of replicates the authors performed (and give the number of replicates performed).

Author’s reply:

We have now added this information to the corresponding figure legends (Figures 1, 2, 3 and 5). In general, all of the depicted gels have been repeated at least 3 times (with the exception of Figures 1C and Figure S3, which has been repeated 2 times) with similar outcomes.

- For key RNA and EMSA gels, it would be nice to see data from additional replicates in the supporting information. This will give readers more confidence in interpreting many of the authors’ exciting data, including the EMSA gel patterns (which point to S4/S5 being the key RNA region for binding).

Author’s reply:

Below, we have pasted results from replicate experiments, as requested by the reviewer. We leave it to the editor whether these should be included in the supporting information.

- For EMSA assays, could the authors speculate what may give rise to the different shift patterns of the S5 mismatch? It does not seem to be a simple loss of binding.

Author's reply:

This is indeed an observation worth mentioning. Since the migration of the S5 target RNA is still affected by the presence of the Cmr complex, we speculate that this might represent partial binding of the more downstream (matching) part(s) of the target RNA (section S6). We did not have conclusive evidence to support this hypothesis and since we do not always observe these different shift patterns as strongly (see replicate), we did not feel comfortable adding this speculation to the manuscript.

- For EMSA assays, please give the concentration of target RNAs used.

Author's reply:

A final concentration of 13.3nM target RNA was used in the EMSA experiment. This has now been added in the methods section at line 655.

- Figure 2F: could the authors discuss why the mismatch at position 6 results in no change in cOA production? Is there an explanation based on the Cmr complex structure? Is it possible the cOA production pattern upon having mismatches at nucleotide 3-5 could recur at downstream positions?

Author's reply:

The mismatch at position 6 does not have any effect on cOA production for the same reason it has no effect on RNA cleavage activity. Our earlier work on the structure of the TtCmr complex (Taylor *et al.*, Science 2015) showed that the thumb-like extension in the Cas7 subunits displaces this particular nucleotide, preventing it from base pairing with the crRNA, similar to the Cascade-like structures of type I systems.

We have discussed this briefly in the discussion section at lines 371-377.

"Our results show that, in contrast to target RNA degradation, cOA production is dependent on base pairing with the first seven 5' nucleotides of the spacer part of the crRNA (with the exception of the nucleotide at position six), a region that we designated as the CAR (Cas10 Activating Region). In agreement with previous work on type I and type III complexes, the nucleotide on the crRNA after each segment (i.e. every sixth nucleotide) is excluded from base pairing with the target RNA, due to the thumb-like extension of Cas7^{23,38}"

Base-pairing propagation all the way towards the 3' end of the protospacer on the target RNA is what triggers cOA production, due to the positioning requirements of the target RNA in the Cas10 subunit. We therefore expect that small point mutations further away from this region would not have a dramatic impact. Indeed, mutating the first nucleotide of segment 2 (position 7) did not dramatically impact cOA production (Figure 2F), whereas the first nucleotide of segment 1 (position 1) did abolish cOA production. However, big disturbances in binding (preventing complete base pairing towards the 3' end of the protospacer), like the complete segment 2 mismatch, do hinder cOA production as shown in our results (Figure 3C, F and I).

For the nucleic acid detection part:

- Please do not refer to methods described in earlier literature (e.g. previously performed LAMP protocols); please just inform readers exactly what was done for the experiments shown in this manuscript.

Author's reply:

We carefully revised the method section and included more details on how to replicate our results. More details on the LAMP protocol have now been added to the methods section on lines 686-690, which now reads:

"The LAMP reaction in the 2-step LAMP-CRISPR detection setup (Figures 5C and 5D) was performed using the WarmStart® LAMP Kit (NEB #E1700), using previously published primers (Table S1) designed for amplifying SARS-

CoV-2 RNA⁴⁵. Final primer concentrations of 0.2 μM, 1.6 μM and 0.8 μM for the F3/B3, FIP/BIP and LoopF/LoopB primers were used respectively.”

- Can the authors provide more rational why among CARF-RNases TTHB144 is selected, beyond it being well-characterized?

Author’s reply:

At this stage we do not know the exact cOA production repertoire (e.g. the number of AMP moieties in the ring-like structure of cOA) of the two native type III complexes (type III-A and -B) in *Thermus thermophilus*. To ensure CARF activation with the cOAs produced by the type III-B Cmr complex, we looked at the three CARF protein (TTHB144, -152 and -155) options we had. We reasoned that TTHB144 would be the best candidate as, 1) TTHB152 is encoded in the type III-A operon and will most likely be compatible with cOAs produced by that system and 2) TTHB155 is described as a DNA nickase, which would not be compatible with the readout we used for SCOPE (i.e. cleavage of a reporter RNA).

We have now better explained this at line 411-415

“For signal amplification purposes and to achieve an easy read-out (RNase activity), we selected TTHB144 - one of the three native CARF proteins present in the genome of *T. thermophilus*^{27,30,65}. While TTHB152 is also characterized as a CARF-RNase, it is encoded in the type III-A operon, suggesting that it would be more compatible with the cOAs produced by type III-A complex.”

- Do the authors use endogenous or reconstituted TtCmr for these assays? It seems the reconstituted TtCmr-46 is used for Figure 5A, but unclear about the rest of Figure 5.

Author’s reply:

We agree that this was not obvious. For all of the assays presented in Figure 5, we used the reconstituted TtCmr-46 complex. To clarify this, we have now included it in the methods section on lines 651 and 667 and in the legend of Figure 5.

- For Figure 5A, please give sequence of the non-target RNA (NT) used as a control.

Author’s reply:

We thank the reviewer for spotting this, the sequence of the non-target RNA used in Figure 5A has now been added to Table S1.

- Figure 5A, B, D, F: I presume the cy5 reporter is used in Figure 5A, while RNase alert is used in 5B, D, and F. But better to write these details out and leave no ambiguity.

Author’s reply:

The Cy5 reporter was indeed used in 5A while the RNaseAlert was used in 5B, D, F and E. This has now been clarified in the methods section on line 684-685.

- For the clinical sample assessment, what is the exact reaction time for the LAMP step? What is the exact reaction time for the CRISPR step? Giving time ranges as in Figure 5C is ok as an overview, but insufficient when trying to associate methods with results.

Author's reply:

The 2-step protocol on the human swap samples was performed with a 30 min pre-amplification LAMP reaction and a 15 min CRISPR detection step. This information has now been added to the methods section, lines 695-697.

- What was the volume input of the clinical RNA extracts the authors used in their reaction?

Author's reply:

In the pre-amplification LAMP reaction, 5 μ L of sample RNA extract was used. This has been added to the methods section, line 696.

- My understanding is the authors prepare the CRISPR mix and add this directly to the LAMP reaction post-amplification. Is this correct? Is the concentration of CRISPR mix components final concentration, after adding to the LAMP reaction?

Author's reply:

The reviewer's assumption is correct, the CRISPR mix is added directly to the pre-amplification reaction. The concentrations given are the final concentrations after addition to the pre-amplification reaction, which has now been clarified more clearly in the methods section, line 698.

- What is the delta-signal SCOPE value shown in Table S2? Are these the blank-subtracted fluorescence similar to what is shown in Figure 5? But why are the numbers from Table S2 and Figure 5 not in the same range?

Author's reply:

The delta-signal SCOPE value is the, blank subtracted, increase in fluorescent signal during the 2nd step (CRISPR detection) of the 2-step protocol (15 mins of incubation). This has been added to the methods section, lines 711-712.

"The delta-signal SCOPE value was calculated by taking the, blank subtracted, increase of fluorescent signal over time during the 15 mins of the CRISPR detection step in the 2-step protocol."

There is indeed a difference between the range of values presented in Figure 5 (panels D and F) and Table S2, the data from Figure 5D and 5F were obtained on a Bio-Rad CFX96 machine while data in Table S2 was obtained on an on-site Applied Biosystems 7500 machine, dictated by the location of the testing facility for patients samples. While identical amounts of fluorescent probe were used in these experiments, the different machines gave rise to the different values ranges, as noted by the reviewer. The type of machine used for each Figure has been included in the methods section, lines 702

- Text associated with Figure 5E: the agreement with RT-PCR is up to Ct 37 based on the data, not to Ct 39 as written in the text

Author's reply:

We thank the reviewer for spotting this typo, "39" has been changed to "37" accordingly, lines 307 and 429.

- Figure 5F: for the one-pot SCOPE result, what is the SCOPE reaction time that gives rise to this result? In addition to poorer LoD, if the fluorescence signal from the one-pot version is consistently lower than the two-put version, the authors should comment so.

Author's reply:

The total reaction time of the one-pot version of the assay was 180 minutes, which is now mentioned in the relevant section of the methods, line 714. We have also expanded the text addressing the current limitations of

our 1-pot setup (lower LOD, longer reaction time), which we ascribe to the lower incubation temperature and potential incompatibility with factors from the LAMP reaction in the discussion on the discussion lines 440-445. *“Although similar reaction component concentrations compared to the 2-step protocol were used, both the LOD (800 aM) and reaction time (180 min) were affected in the one-pot approach. This is likely due to the sub-optimal incubation temperature of 55°C combined with potential interfering reaction components to the LAMP reaction. Optimization consisting of changing TTHB144 and reporter RNA concentrations, for example, could improve this issue.”*

- Figure 5A: there is a mentioning of testing a mismatched sequence in segment 6 for TTHB144 activation in the figure legend, but the result is absent

Author’s reply:

We thank the reviewer for spotting this piece of left-over text in the figure legend. Originally, we included S6 in the figure, but we opted to remove this result since it did not add anything to the conclusions from this part of the manuscript. The figure legend had been changed accordingly. We have now included 2 additional replicates of these results and have included the S6 results in these figures in case the reviewer is curious. As expected, the S6 target did only mildly affect TTHB144 activity.

Reviewer #2 (Remarks to the Author):

The authors describe the development of proof-of-concept diagnostic based on a type III CRISPR-cas platform, SCOPE. A majority of the manuscript focuses on the study of type III CRISPR Cas RNA binding and cleavage mechanisms, and data on diagnostic application is presented for the detection of SARS-CoV-2. Authors report a class 1-based CRISPR-cas nucleic acid detection tool, claiming high specificity and rapid detection. Authors demonstrate fluorescence-based SARS-CoV-2 detection in buffer, detecting 1 nM of target RNA after a few seconds of incubation. In addition, a 2-step protocol using (RT-) LAMP preamplification is described and authors report attomolar limits of detection. This 2-step protocol was tested in 80 human nasal swabs in correlation with PCR measurements. Using a CT threshold of 39, 60 samples were positive by PCR and all 60 were also positive by the SCOPE test. Lastly, authors demonstrate a one-pot synthesis protocol for SARS-CoV-2 detection, reporting a limit of detection of 800 aM. The manuscript is well-written, with most of the work being an advance that can be special interest to the CRISPR community. Data presented for SCOPE as a diagnostic is intriguing and limits of detection are impressive, but evaluation of test sensitivity is insufficient. CRISPR-based SARS-CoV-2 tests are potential tools to combat the pandemic, however, if the following manuscript is to present SCOPE as a diagnostic tool, rigorous evaluation of test performance and details of test methods should be added to strengthen the claim.

Author’s reply:

We thank the reviewer for their positive words about our work. As acknowledged by the reviewer, the focus of the study is indeed on the new fundamental insights on the modus operandi of type III CRISPR-Cas systems (5 out of the 6 figures), that formed the basis for the development of SCOPE. With SCOPE, we provided a proof-of-concept that type III CRISPR-Cas systems have a couple of useful characteristics for diagnostic purposes. Akin to other CRISPR detection platforms (e.g. DETECTR and SHERLOCK), we would like to improve SCOPE in the future by improving the throughput and sensitivity of our tool even further (i.e. developing a 1-pot reaction with similar reaction kinetics and LOD as the here presented 2-pot reaction). We therefore agree that if we were to present SCOPE as a validated diagnostic tool some more work is certainly required (for example, by testing SARS-CoV-2 variants to test its discriminative power as suggested by the reviewer). Lastly, we appreciate the questions of the reviewer on the lack of details in the methods, which were raised by reviewer #1 as well. We have taken these comments as a starting point to revise this section thoroughly (also see our replies to reviewer #1 on this matter).

Comments:

1. LOD for the fluorescence-based test, the 2-step test, and the one-pot reaction test are stated but there is no description of how LODs were determined. Clearly state how the LOD was calculated. Include description of replicates for measurements.

Author's reply:

The LOD was determined by the final target RNA concentration that was present in the reaction mixture in Figure 5B & 5F, which resulted in a significant increase in fluorescence compared to the non-target RNA control. For Figure 5D, the final target concentration in step 1 (RT-LAMP step) was taken to calculate the LOD. Figure 5B, D and F were performed in triplicate (included in the revised manuscript). This information was added to the methods section and figure legends.

"The limit of detection (LOD) was determined by the lowest concentration of target RNA in the reaction mixture in step 1, which led to a significant increase in fluorescence compared to the non-target RNA control."

2. How was SCOPE signal defined? SI shows delta signal SCOPE, how was this calculated?

Author's reply:

This issue was rightfully brought up by reviewer #1 as well. The delta-signal SCOPE value is the (blank-subtracted) increase in fluorescent signal during the 2nd step (CRISPR detection) of the 2-step protocol (15 mins of incubation). This has been added to the methods section, lines 711-712. Please also see our reply to a similar comment from reviewer #1.

3. Authors demonstrated specificity via denaturing PAGE and should perform similar control experiments to validate diagnostic tests. For example, include fluorescence signal generated after incubation for other corona viruses such as hCov NL63, hCov OC43, and hCov 229E, showing data for fluorescence signal vs incubation time and signal at the final incubation time.

Author's reply:

The healthy control patients included in Figure 5E are essentially a more complex version of the "non-target RNAs" controls that we used for the PAGE analysis, which in both cases gave no signal or cleavage respectively. We agree with the reviewer that it would be interesting to test the SARS-CoV-2 variants the reviewer suggested (to see if SCOPE can discriminate between more subtle changes in the protospacer, as we did with the segment and single point mutations in the PAGE analysis). On top of the efforts required to collect and get consent for these samples, the SCOPE tool presented in this study is a proof-of-concept type III CRISPR-Cas based nucleic acid detection method and is presented as such. More thorough investigation into the diagnostic characteristics is something we would like to do in the near future.

4. What is the cut off value to determine positive and negative nasal swabs by SCOPE? What was the criteria for determining this cutoff?

Author's reply:

Our current SCOPE tool results in a binary signal, either the signal goes up high or stays around zero, leaving little ambiguity. The graph below depicts the Delta SCOPE signal for all samples tested (based on the values presented in Table S2), a clear distinction can be seen between a positive result and negative result. However, a cut-off of 2 standard deviations, from the average of all qPCR-scored positive samples, was applied. This clarification has been added to the method section, lines 715-171. Furthermore, a graph depicting the Delta Scope values, showing the binary nature of signals obtained, has been added to the supplementary information (Figure S5).

“For the human swap samples presented in Figure 5E, a sample was considered positive if the average delta-SCOPE value was within 2 standard deviations of all qPCR-scored positive samples.”

5. Provide brief description of where nasal swabs were obtained from, patients? Healthy adults? COVID19 patients? Were replicate measurements in nasal swabs measured?

Author’s reply:

Combined human throat/nasopharyngeal swab samples were obtained at a community testing center (Utrecht, the Netherlands) from adults with either SARS-CoV2- related symptoms or with contacts of infected persons. As samples were collected within the framework of the national COVID measures and testing efforts, samples were routinely collected at interval timepoints. This information has now been revised in the materials & methods section at lines 668-672

Due to the limit amount of patient material, both the PCR and SCOPE analyses has been performed once for each sample.

6. Authors should briefly discuss advantages and limitations of SCOPE for diagnostic applications including over all test time, sample volumes needed, sample throughput, etc.

Author’s reply:

The closing words have been revised to contain the extra discussion points brought up by the reviewer.

“A couple of CRISPR-based nucleic acids detection platforms have been developed over the last years, such as the Class2-based DETECTR and SHERLOCK platforms^{66,67}. However, SCOPE is the first Class 1-based CRISPR-Cas nucleic acid detection tool, with some very useful characteristics: highly sensitive (requiring very little sample input, see methods) and specific (flexible seed, stringent CAR), quick (detection within seconds), flexible (PAM-independent guide design), highly robust (long shelf-life). Furthermore, the potential for a 1-pot system in combination with RT-LAMP offers opportunities to increase the throughput of our tool. The thermophilic nature of the system potentially means that the system is less affected by, enzyme-mediated, inhibitory factors in crudely extracted samples. While the 2-step approach currently limits efficient high-throughput testing due to the extra (manual) step that is required, an efficient 1-pot system would mitigate this. The use of a standard fluorophore and no proprietary plastics further facilitates integration in current high throughput testing facilities. All these favorable features make the SCOPE approach a highly attractive alternative over the currently available detection tools.”

Reviewers' Comments:

Reviewer #1:

Remarks to the Author:

The authors have fully addressed my and the other reviewer's concerns. I recommend publication in due course. Re: the replicate experiments, there is rich information in there so I suggest the authors incorporate them as parts of the supporting information.

Reviewer #2:

Remarks to the Author:

Authors thoroughly addressed all comments. After authors address a minor revision (see below), this manuscript is suitable for publication in Nature Communications.

Minor revisions:

With regards to comment: 5. Provide brief description of where nasal swabs were obtained from, patients? Healthy adults? COVID19 patients? Were replicate measurements in nasal swabs measured?

Author's reply: Combined human throat/nasopharyngeal swab samples were obtained at a community testing center (Utrecht, the Netherlands) from adults with either SARS-CoV2- related symptoms or with contacts of infected persons. As samples were collected within the framework of the national COVID measures and testing efforts, samples were routinely collected at interval timepoints. This information has now been revised in the materials & methods section at lines 668-672 Due to the limit amount of patient material, both the PCR and SCOPE analyses has been performed once for each sample.

Authors should include in the text that data for human nasal swabs was obtained from one replicate measurement.

REVIEWERS' COMMENTS

Reviewer #1 (Remarks to the Author):

The authors have fully addressed my and the other reviewer's concerns. I recommend publication in due course. Re: the replicate experiments, there is rich information in there so I suggest the authors incorporate them as parts of the supporting information.

Authors' reply:

We have now added all the replicate experiments to the supporting information (Figure S6).

Reviewer #2 (Remarks to the Author):

Authors thoroughly addressed all comments. After authors address a minor revision (see below), this manuscript is suitable for publication in Nature Communications.

Minor revisions:

Authors should include in the text that data for human nasal swabs was obtained from one replicate measurement.

Authors' reply:

We have now added this information to the legend of the corresponding figure (Figure 5E)